

# Mycorrhization of *Quercus acutissima* with Chinese black truffle significantly altered the host physiology and root-associated microbiomes

Xiaoping Zhang[1,2,*], Lei Ye[1,*], Zongjing Kang[1,2], Jie Zou[1,2], Xiaoping Zhang[2] and Xiaolin Li[1]

[1] Soil and Fertilizer Institute, Sichuan Academy of Agricultural Sciences, Chengdu, China
[2] Department of Microbiology, College of Resources, Sichuan Agricultural University, Chengdu, China
[*] These authors contributed equally to this work.

## ABSTRACT

**Background**. Our aim was to explore how the ectomycorrhizae of an indigenous tree, *Quercus acutissima,* with a commercial truffle, Chinese black truffle (*Tuber indicum*), affects the host plant physiology and shapes the associated microbial communities in the surrounding environment during the early stage of symbiosis.

**Methods**. To achieve this, changes in root morphology and microscopic characteristics, plant physiology indices, and the rhizosphere soil properties were investigated when six-month-old ectomycorrhizae were synthesized. Meanwhile, next-generation sequencing technology was used to analyze the bacterial and fungal communities in the root endosphere and rhizosphere soil inoculated with *T. indicum* or not.

**Results**. The results showed that colonization by *T. indicum* significantly improved the activity of superoxide dismutase in roots but significantly decreased the root activity. The biomass, leaf chlorophyll content and root peroxidase activity did not obviously differ. Ectomycorrhization of *Q. acutissima* with *T. indicum* affected the characteristics of the rhizosphere soil, improving the content of organic matter, total nitrogen, total phosphorus and available nitrogen. The bacterial and fungal community composition in the root endosphere and rhizosphere soil was altered by *T. indicum* colonization, as was the community richness and diversity. The dominant bacteria in all the samples were Proteobacteria and Actinobacteria, and the dominant fungi were Eukaryota_norank, Ascomycota, and Mucoromycota. Some bacterial communities, such as *Streptomyces, SM1A02,* and *Rhizomicrobium* were more abundant in the ectomycorrhizae or ectomycorrhizosphere soil. *Tuber* was the second-most abundant fungal genus, and *Fusarium* was present at lower amounts in the inoculated samples.

**Discussion**. Overall, the symbiotic relationship between *Q. acutissima* and *T. indicum* had an obvious effect on host plant physiology, soil properties, and microbial community composition in the root endosphere and rhizosphere soil, which could improve our understanding of the symbiotic relationship between *Q. acutissima* and *T. indicum,* and may contribute to the cultivation of truffle.

Corresponding authors
Xiaoping Zhang,
aumdwsb@sicau.edu.cn
Xiaolin Li, kerrylee_tw@sina.com

## INTRODUCTION

*T. indicum*, an Asian black truffle that is phylogenetically and morphologically related to Périgord black truffle (*T. melanosporum* Vittad), is the major commercial truffle species in China and is a highly economically valued edible fungus (*Liu et al., 2011*). Since its discovery and export to Europe in the 1990s, *T. indicum* has attracted increasing attention (*Geng et al., 2009*). As ectomycorrhizal fungi characterized by hypogeous fruiting bodies, truffles need to infect host plants and develop a symbiosis with them to complete their lifecycle (*Kües & Martin, 2011*; *Healy et al., 2016*). At present, truffle cultivation mainly focuses on synthesizing truffle-colonized seedlings and establishing truffle plantations (*Liu et al., 2011*). *T. indicum* has a wide adaptability to host plants, and it's specificity of colonization is not absolute. It can establish symbiotic associations with the roots of *Pinus armandii, Cyclobalanopsis glauca, Quercus aliena, Populus bonatii, Carya illinoinensis,* and *Corylus avellana,* to name a few, which includes Chinese trees, North American trees, and European trees (*Hu, 2004*; *Bonito et al., 2011*; *Deng, Yu & Liu, 2014*). *Q. acutissima* is an indigenous Chinese broad leaf tree that is widely distributed in China. At present, successful mycorrhizal synthesis with *T. indicum* has been observed for *Q. acutissima* (*Hu, 2004*).

As ectomycorrhizal fungi, truffles can maintain the stability and health of ecosystems in the natural environment to a certain extent, which has important ecological value (*Liu et al., 2011*). During the life cycle of truffles, from mycelium germination to the harvesting of the fruiting bodies, some volatile metabolites are released, and these metabolites can inhibit the germination of the surrounding seeds and other biological communities, creating a brûlé area (*Streiblová, Gryndlerová & Gryndler, 2012*). Some reports showed that bacteria and fungi diversity in the brûlé area is lower, and thus truffles are inferred to affect the associated soil microbial communities (*Mello et al., 2013*; *Zampieri et al., 2016*). Although the microbial community in the surrounding environment varied with different truffles species, truffle growth periods, and different regions and seasons, they were found important to the formation of ectomycorrhizae and truffle ascocarps, as well as their aroma (*Barbieri et al., 2007*; *Splivallo et al., 2015*; *Benucci & Bonito, 2016*; *Deveau et al., 2016*). Some bacteria in rhizosphere soil were screened and confirmed to benefit the synthesis of ectomycorrhizae, referred to as "mycorrhization helper bacteria" (*Wang et al., 2015*; *Fu et al., 2016*). In addition, a diverse microbial community consisting of bacteria, yeast, and filamentous fungi was found in the ectomycorrhizae and fruiting bodies of truffles, many of which are endophytes (*Zhou & Wei, 2013*; *Splivallo et al., 2015*). Endophytes can produce bioactive compounds, participate in the host metabolic processes, and play important roles in the growth and fitness of plants (*Hardoim et al., 2015*; *Ludwig-Müller, 2015*). These microorganisms are closely associated with truffle growth and their interaction with truffles may be key to the artificial cultivation that is currently lacking. Additionally, the colonization of ectomycorrhizal fungi with root system can improve the soil structure, and the soil characteristics also influence truffle ectomycorrhizae development and sporocarp formation (*Alonso et al., 2014*). Furthermore, ectomycorrhizal fungi can improve water absorption and nutrient utilization by host plants, which is beneficial to plant growth. Following ectomycorrhizal formation, the expression of some defense genes induced by

plants enhances the biotic and abiotic stress resistance of host plants, including drought resistance, disease resistance, and pest resistance (*Kennedy, 2010*; *Lilleskova, Hobbieb & Horton, 2011*).

Although truffles have been studied for many years, their complex growth mechanisms are not well understood and successful artificial cultivation has not been achieved. Previous studies focused greatly on the mycorrhization of seedlings with truffles and the microbial communities associated with truffles, including microbes in the soil and ascocarps. However, how these microbial communities play their roles in truffle formation, and the relationship and interaction between the ectomycorrhizae, microorganisms, and host plants, is unclear. We thus synthetized the ectomycorrhizae of *Q. acutissima* with *T. indicum* to explore the effect of *T. indicum* colonization on the host plant. Changes in root morphology and physiology, and the rhizosphere soil properties were also detected. In addition, we explored the influence of the symbiotic relationship of *Q. acutissima* and *T. indicum* on bacterial and fungal community composition in the root endosphere and rhizosphere soil using high-throughput sequencing. We hypothesized that the truffles colonization could alter the host plant physiology, and the microbiome in rhizosphere soil and root endosphere would be shaped in response to the truffle ectomycorrhization, which could contribute to reveal the ecological mechanisms of truffle during the early symbiotic stages, with the aim of providing a foundation for the artificial cultivation of truffles.

## MATERIAL AND METHODS

### Cultivation of *Q. acutissima* seedlings and truffle inoculation treatment

The *Q. acutissima* seeds were obtained from Taian city, Shandong province, China. The seeds were washed with water to remove impurities and then soaked in 0.5% potassium permanganate solution for 2 h. Following this, the seeds were surface-sterilized with ultra-pure water and then sowed in nursery substrate, as detailed in a previous report (*Li et al., 2017*). After 2 months, the seedlings that exhibited good growth and were of similar height were selected for the root-tip cutting treatment. Root-tip cutting is an efficient technique for plant growth management (*Shabala et al., 2009*). Approximately 1 cm of root tip from the taproots was removed with sterilized scissors. The *Q. acutissima* seedlings that had their roots-tips removed were then respectively transplanted into a plastic container (about 155 mL) with sterilized cultivation substrate. The cultivation substrate (pH 7.5) was composed of peat, organic soil (from Yuexi County in Sichuan, China), vermiculite, and water at a volume ratio of 1:1:1:1.5. Inoculation was performed at the same time that the plants were transplanted.

*T. indicum* was purchased from Huidong County, China. The ascocarps, which had been surface disinfected with 75% alcohol, were pulverized and blended to obtain the spore powder (*Wan et al., 2015*). Two grams of spore powder was inoculated into the cultivation substrate. The uninoculated *Q. acutissima* seedlings were used as controls. The number of inoculated and uninoculated *Q. acutissima* seedlings (controls) was both sixty. Three biological replicates were set in each treatment, so each bioreplicate had 20 seedlings. All
seedlings were cultivated in greenhouses under the same conditions (Fig. S1). During the cultivation period, plants were irrigated with tap water every 2 or 3 d.

## Sample collection and analysis

Six months after inoculation, the plant root tips and the rhizosphere soil in the cultivation substrate were collected. Briefly, take the seedlings (mycorrhization with *T. indicum* and uninoculated seedlings) out of the substrate and collect the rhizosphere soils with sterilized gloves. The soils used for high-throughput sequencing were stored in 2ml EP tubes in −80 °C refrigerator and the soils used for properties determination were dried and then stored in plastic bag at room temperature. The properties of the ectomycorrhizosphere soil (soil around the roots mycorrhization with *T. indicum*) and rhizosphere soil (soil around the roots of uninoculated plants) samples were measured according to the previous method (*Li et al., 2016*), including pH, organic matter (OM), total nitrogen (TN), available nitrogen (AN), total phosphorus (TP), available phosphorus (AP), total potassium (TK), available potassium (AK), available calcium (ACa), and available magnesium (AMg). As for *Q. acutissima* seedlings, clean the roots with water and the morphological analysis of the ectomycorrhizae from *Q. acutissima* roots was performed by microscope. The number of root segments colonized by *T. indicum* was counted according to the mycorrhizal fungal structures, with a total of 30 root segments randomly selected and observed for each seedling (*Andrés-Alpuente et al., 2014*). The mycorrhizal colonization rate was calculated from this, expressed as: mycorrhizal colonization rate (%) = (root segments colonized by *T. indicum* / total root segments observed) ×100. After identifying the mycorrhizae, the *Q. acutissima* roots, including two treatments (roots colonized with truffle mycelia and uncolonized roots) were surface disinfected, after which their DNA was extracted for high-throughput sequencing. Each treatment had three repetitions and each repetition was at least 1 g. The ectomycorrhizae of *Q. acutissima* with *T. indicum* were assigned to "A," and the ectomycorrhizosphere soil was assigned to "B". Roots from the control plants that were not colonized with *T. indicum* belonged to "D," and the rhizosphere soil belonged to "C". All samples had three replicates.

## Determination of plant physiological indices

The root activity, biomass, chlorophyll content in leaves, activities of superoxide dismutases (SOD) and peroxidase (POD) in roots were measured. The root activity determination was based on triphenyl tetrazolium chloride (TTC) method described by *Zhang et al. (2013)*. Briefly, put 0.5 g fresh root in the 10 mL solution mixed with 5 mL 0.4% TTC solution and equal volumes of phosphate buffer. The mixed solution was then kept in oven at 37 °C for 2 h. Then add 2 mL 1 mol/L $H_2SO_4$ to the mixed solution. Dry the root and grind roots fully with ethyl acetate in a mortar, finally with ethyl acetate to 10 mL. The liquid was measured at the 485 nm of a spectrophotometer to get the absorbance. Root activity = the amount of TTC reduction (µg)/fresh root weight (g) × time (min).

The biomass of each seedling was assessed based on the aboveground and belowground dry matter weight. For dry weight determination, the seedlings were first washed to remove any substrate attached to the roots, after which they were oven-dried at 105 °C for 30 min

to halt respiration, and then oven-dried at 75 °C to achieve constant weight (*Moore et al., 2002*). The dry weight of aboveground and belowground parts was measured once the material had cooled.

Chlorophyll content was measured spectrophotometrically using fresh leaf samples without main vein. The same sections of the leaves at the center of *Q. acutissima* seedlings were selected and determined according to previous method (*Taïbi et al., 2016*). Briefly, slice the leaves and grind them fully with 80% acetone (v/v). The extraction was filtered and the volume of it was added to 10 mL with acetone. The absorbance of the extraction was recorded at 663 and 645 nm for calculations of chlorophyll content. Chlorophyll content expressed with (mg)/fresh leaf weight (g).

SOD and POD are antioxidant enzymes. The SOD activity was determined according to its ability to inhibit the photochemical reduction of nitroblue tetrazolium (NBT), following the method of Fridovich (*Fridovich, 2011*). POD activity was measured using the guaiacol method with the theory that $H_2O_2$ can oxidize the guaiacol under the catalysis of peroxidase, following the method of Meloni (*Meloni et al., 2003*).

All the determinations were repeated at least three times.

## DNA extraction, PCR amplification, and HiSeq sequencing

DNA of the tissues and endophytes in the roots was extracted with the hexadecyl trimethyl ammonium bromide (CTAB) method (*Li et al., 2017*). DNA of the rhizosphere soil samples was extracted using a MoBio PowerSoil® DNA Isolation Kit. The extracted DNA was detected by 1% agarose gel electrophoresis.

The 16S V4 and ITS1 region amplification of all samples was respectively performed by the universal primers 515F (5′-GTGCCAGCMGCCGCGGTAA-3′) - 806R (5′-GGACTACH VGGGTWTCTAAT-3′) and ITS1F (5′-CTTGGTCATTTAGAGGAAGTAA-3′) - ITS2 (5′- GCTGCGTTCTTCATCGATGC-3′) with the barcode on ABI GeneAmp® 9700 PCR instrument (*Caporaso et al., 2012*; *Huang et al., 2015*). After mixing and detecting with 2% agarose gel electrophoresis, the PCR products were recycled and purified with Agencourt AMPure Beads (Beckman Coulter, Indianapolis, IN). Finally, the PCR products were quantified using Quant-iT PicoGreen dsDNA Assay Kit and then mixed in proportion according to the concentration of each sample.

PCR amplicon libraries were generated by an Illumina TruSeq Nano DNA LT Sample Prep Kit (FC-121-4001). The library quality was then assessed and amplicon sequencing was performed on an Illumina HiSeq 2500 platform, generating 2×300 bp sequences. All raw data were submitted to the Sequence Read Archive (SRA) database with the accession numbers SRR7791517–SRR7791532.

## Sequence data processing and statistical analysis

The raw data were firstly saved in FASTQ format. The obtained reads were firstly spliced according to the overlap relation and then quality-controlled and filtered. Trimmomatic, FLASH, Usearch, and QIIME software were used for these processes (*Caporaso et al., 2010*; *Magoč & Salzberg, 2011*). High-quality sequences more than 97% similarity were classified into an OTU with UCLUST (*Edgar, 2010*). To obtain the classification information for each

OTU species, the RDP classifier was used for the taxonomic analysis of OTU representative sequences based on the SILVA (Release 119; http://www.arb-silva.de) and Unite (Release 6.0; http://unite.ut.ee/index.php) databases. Rarefaction curves were used to estimate coverage. The alpha-diversity indices including Chao1, Shannon, and Simpson were analyzed with QIIME (version 1.7.0). The beta diversity which indicates the differences of microbial communities among the samples was reflected by Non-metric multidimensional scaling (NMDS). Meanwhile, the ANalysis Of SIMilarity (ANOSIM) test was used to test significant differences between the treatments. The shared OTUs were presented in a Venn diagram. Heatmaps were drawn using R software (R Core Team, 2014) in order to cluster and analyze the more abundant phyla and genera in samples and to evaluate the taxonomic composition of the microbial communities. LEfSe (linear discriminant analysis, LDA) was performed to reveal the taxa of microbial communities that had differential abundance in the different treatments at all taxonomic levels.

One-way analysis of variance (ANOVA) and independent sample $t$-tests were performed in in SPSS v21.0 (IBM Inc., Armonk, NY, USA). T-tests were used for the analysis of soil properties and plant physiological indices. The least significant difference (LSD) was performed to determine the significant pairwise differences between different treatments. All significant differences were assessed at $P < 0.05$.

## RESULTS

### The effect of *T. indicum* inoculation on *Q. acutissima*

Six months after inoculation of *T. indicum,* mycorrhization was successfully detected in the inoculated *Q. acutissima* seedlings, and the calculated mycorrhizal colonization rate was $52.07\% \pm 13.52\%$. The seedlings that were not inoculated with truffle spores exhibited obvious differences in their root systems based on morphological analysis (Figs. 1 and 2). It was evident that the synthesized ectomycorrhizae were monopodial or binary branched and were yellowish-brown.

The root activity, biomass, chlorophyll content in leaves, and SOD and POD activity in the roots of *Q. acutissima* are shown in Table 1. POD activity, biomass and chlorophyll content did not differ significantly between the inoculated and control seedlings. SOD activity increased significantly in the *Q. acutissima* seedlings following inoculation with *T. indicum* compared with the control seedlings ($P < 0.05$). However, the root activity was significantly lower in the *Q. acutissima* seedlings colonized by *T. indicum* ($P < 0.05$).

### Soil properties analysis

Some physicochemical properties of the soil samples collected from the *Q. acutissima* roots are presented in Table 2, including those associated with both the inoculated and uninoculated treatments. With the exception of total potassium and available magnesium, the remaining parameters (including pH, organic matter, total nitrogen, total phosphorus, available nitrogen, available phosphorus, available potassium, and available calcium) all differed significantly between the ectomycorrhizosphere soil and rhizosphere soil (control treatment) ($P < 0.05$). The content of total phosphorus was significantly higher in the ectomycorrhizosphere soil ($P < 0.05$), increasing by 9% compared with that in

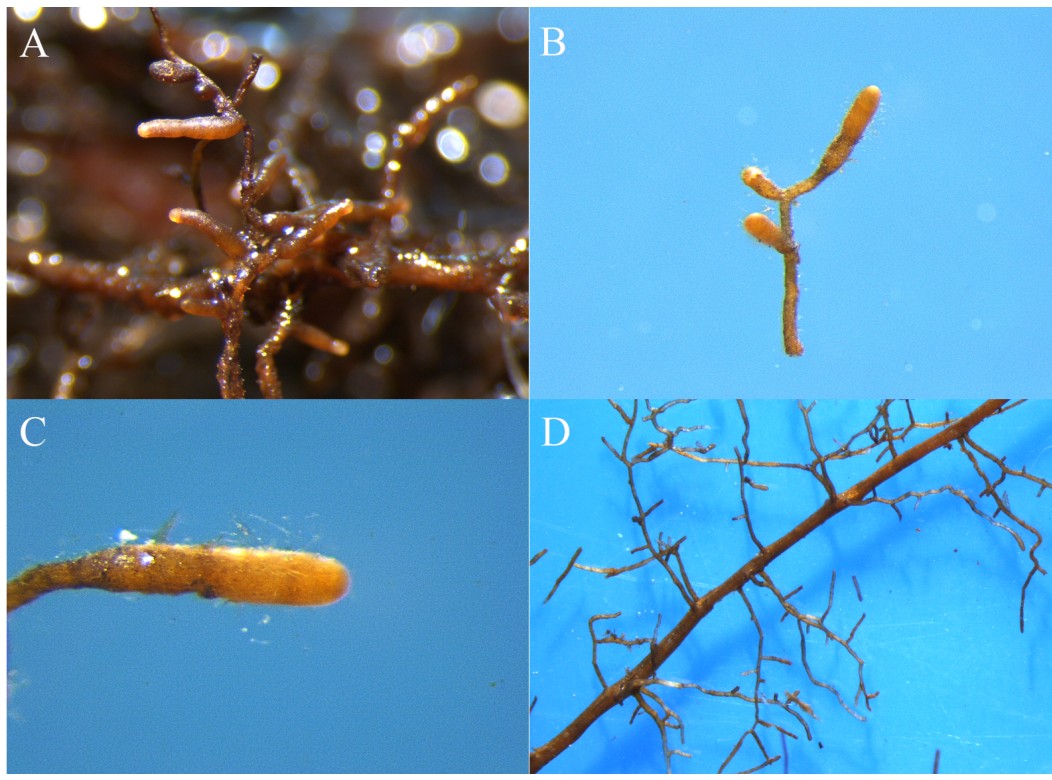

**Figure 1** The ectomycorrhizae of *Q. acutissima* with *T. indicum* (A, B, C) and the root tips of *Q. acutissima* that were not colonized by *T. indicum* (D). Photo credit: Lei Ye.

rhizosphere soil, while the available phosphorus was significantly lower, decreasing by 11% ($P < 0.05$). The content of organic matter, total nitrogen, and available nitrogen were all also significantly higher in the ectomycorrhizosphere soil than the rhizosphere soil ($P < 0.05$), increasing by 3%, 1% and 6%, respectively. Inoculation with *T. indicum* thus improved the carbon and nitrogen levels in the rhizosphere soil. However, the rhizosphere soil had significantly higher pH, available potassium, and available calcium ($P < 0.05$).

## Alpha diversity of bacterial community

A total of 323,364 high-quality sequences were obtained from the 12 samples after quality control procedures, and there were 18,333–33,288 high-quality sequences in each sample (Fig. S2A). The sequences from each sample were classified into 578–1,344 OTUs. The Venn diagram displayed the degree of overlap of the bacterial OTUs among the samples of the four treatments, and 72, 79, 298, and 109 unique OTUs were identified in each treatment, respectively (Fig. 3A). A total of 30 phyla, 88 classes, 148 orders, 275 families, and 522 genera of bacteria and archaea were detected.

The bacterial alpha diversity indices, including observed species (OTU), Chao1, Shannon, and Simpson, are shown in Table 3A. The number of observed species and the Chao1 index, which indicate the community richness, were significantly higher in the uninoculated rhizosphere soil than in the other samples ($P < 0.05$), indicating that

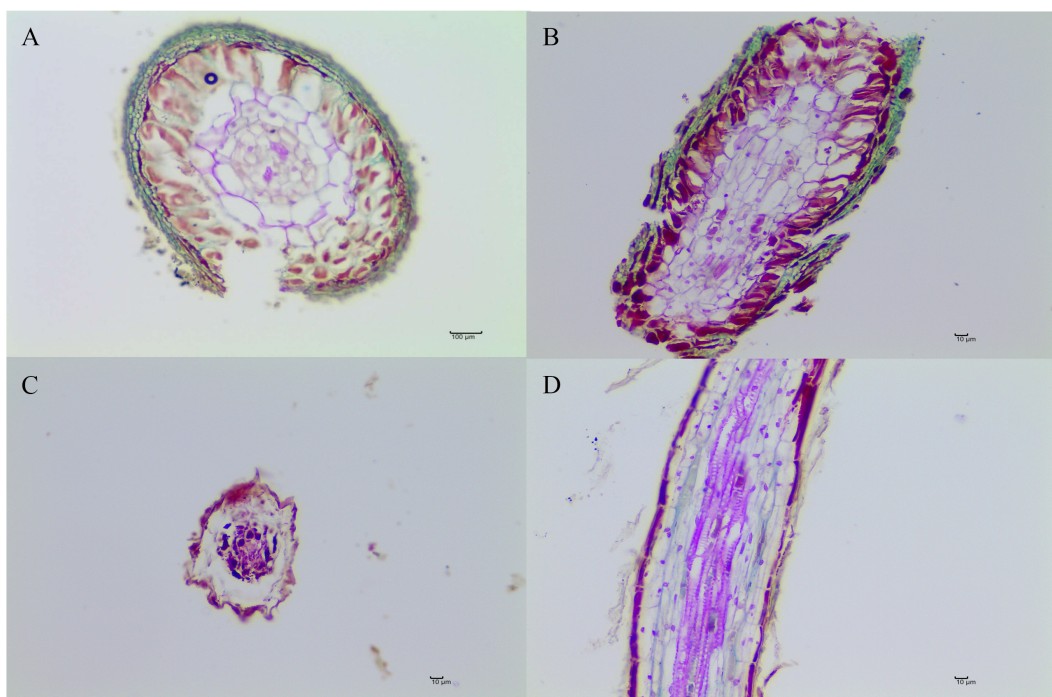

**Figure 2** **The morphological characteristics of *T. indicum* ectomycorrhizae (A, B) and control roots tips that were not colonized by *T. indicum* (C, D).** A and C, transversal sections; B and D, longitudinal sections.

**Table 1** **The root activity, biomass, leaf chlorophyll content, root superoxide dismutases (SOD) activity and root peroxidase (POD) activity of *Q. acutissima* with *T. indicum* mycorrhization or not.** Ectomycorrhizae, ectomycorrhizae from *Q. acutissima* mycorrhized with *T. indicum*. Control roots, roots from cultivated *Q. acutissima* without *T. indicum* colonization. "FW" means fresh weight of the root. Value followed "*" symbol indicates a significant difference between samples ($P < 0.05$). Each value is the mean of three replicates ($\pm$SD).

| Samples | POD activity U/(g*min) FW | SOD activity U/(g*h) FW | Root activity ug/(g*min) FW | Biomass (g) | Chlorophyll mg/g FW |
|---|---|---|---|---|---|
| Ectomycorrhizae | 14.93 ± 6.99 | 102.47 ± 2.58 | 26.25 ± 5.16 | 3.02 ± 0.81 | 28.62 ± 5.43 |
| Control roots | 17.20 ± 6.15 | 68.25 ± 1.77* | 40.95 ± 4.27* | 2.92 ± 0.32 | 32.48 ± 3.68 |

*T. indicum* colonization significantly decreased the bacteria community richness in the rhizosphere soil of *Q. acutissima*. The observed species and Chao1 were lower in the ectomycorrhizae than in the control roots, but did not reach a significant level. The Shannon and Simpson indices represent the community diversity, the greater the Shannon value, the higher the community diversity, while the larger the Simpson index, the lower the community diversity. The ectomycorrhizae had a significantly higher Shannon index ($P < 0.05$) and significantly lower Simpson index ($P < 0.05$), illustrating a higher bacterial diversity than the control roots. The Shannon index was highest in the uninoculated rhizosphere soil samples, while the Simpson index was lowest, indicating that inoculation with *T. indicum* could significantly decrease the bacterial diversity in the rhizosphere soil of

Zhang et al. (2019), *PeerJ*, DOI 10.7717/peerj.6421

**Table 2** **Physical and chemical properties of *Q. acutissima* rhizosphere soil and ectomycorrhizosphere soil.**

| Samples | pH | OM g/kg | TN g/kg | TP g/kg | TK g/kg | AN mg/kg | AP mg/kg | AK mg/kg | ACa cmol (1/2Ca$^{2+}$)/kg | AMg cmol (1/2Mg$^{2+}$)/kg |
|---|---|---|---|---|---|---|---|---|---|---|
| Ectomycorrhizosphere soil | $8.27 \pm 0.01$ | $23.27 \pm 0.21$ | $0.610 \pm 0.00$ | $1.27 \pm 0.05$ | $22.21 \pm 0.42$ | $388.00 \pm 1.73$ | $72.23 \pm 1.29$ | $652.67 \pm 4.72$ | $39.37 \pm 0.93$ | $7.77 \pm 0.29$ |
| Control soil (rhizosphere soil without *T. indicum* associations) | $8.54 \pm 0.04^*$ | $22.53 \pm 0.12^*$ | $0.602 \pm 0.00^*$ | $1.16 \pm 0.02^*$ | $22.17 \pm 0.55$ | $364.33 \pm 2.52^*$ | $80.50 \pm 1.73^*$ | $676.33 \pm 3.21^*$ | $59.27 \pm 0.65^*$ | $7.77 \pm 0.21$ |

**Notes.**

OM, organic matter; TN, total nitrogen; TP, total phosphorus; TK, total potassium; AN, available nitrogen; AP, available phosphorus; AK, available potassium; ACa, available calcium; AMg, available magnesium.

Value followed "*" symbol indicates a significant difference between samples ($P < 0.05$). Each value is the mean of three replicates ($\pm$SD).

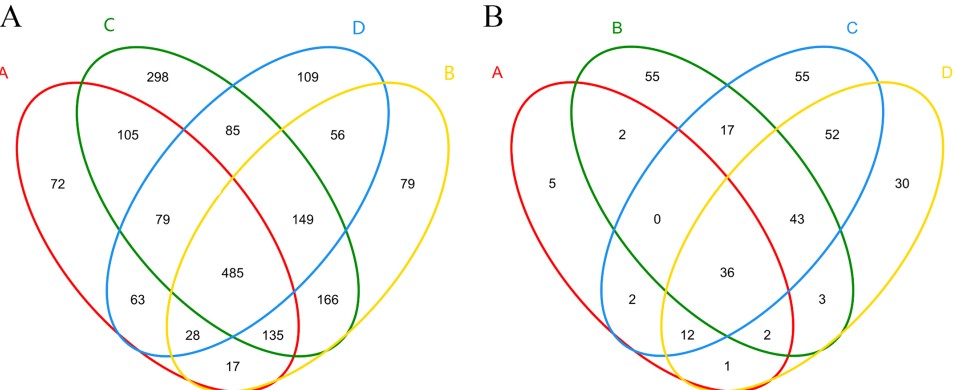

**Figure 3 Shared and unique bacterial (A) and fungal (B) OTUs among samples in different treatments.**
(A), ectomycorrhizae from *Q. acutissima* mycorrhized with *T. indicum*. (B), the ectomycorrhizosphere soil of *Q. acutissima*. (C), the rhizosphere soil without *T. indicum* associations (control soil). (D), roots from cultivated *Q. acutissima* without *T. indicum* colonization (control roots).

**Table 3 Community richness and diversity indices of bacteria (A) and fungus (B) associated with *Q. acutissima* roots and rhizosphere soils with or without *T. indicum* mycorrhization.**

**(A)**

| Samples | OUT | Chao1 | Shannon | Simpson | Coverage |
|---|---|---|---|---|---|
| A | 759.33 ± 94.48a | 938 ± 122.44a | 3.06 ± 0.43a | 0.20 ± 0.07a | 0.9923 ± 0.0011a |
| B | 908.67 ± 26.39a | 1,101 ± 47.84a | 5.21 ± 0.07b | 0.02 ± 0.00b | 0.9895 ± 0.0004b |
| C | 1,333.67 ± 11.06b | 1,464.33 ± 10.97b | 5.81 ± 0.08c | 0.01 ± 0.00b | 0.9923 ± 0.0005a |
| D | 767.33 ± 169.51a | 997 ± 228.68a | 2.32 ± 0.20d | 0.36 ± 0.03c | 0.9923 ± 0.0022a |

**(B)**

| Samples | OUT | Chao | Shannon | Simpson | Coverage |
|---|---|---|---|---|---|
| A | 38.33 ± 2.31a | 58.33 ± 24.91a | 0.42 ± 0.08a | 0.84 ± 0.03a | 0.9997 ± 0.0006a |
| B | 118.33 ± 6.66b | 140.33 ± 5.13b | 0.91 ± 0.47a | 0.74 ± 0.16a | 0.9995 ± 0.0035ab |
| C | 176 ± 5c | 192.67 ± 5.86c | 2.89 ± 0.42b | 0.14 ± 0.07b | 0.9993 ± 0.0006bc |
| D | 143.33 ± 7.23d | 175 ± 20.81c | 2.02 ± 0.19c | 0.27 ± 0.08b | 0.9991 ± 0.0020c |

**Notes.**
A, ectomycorrhizae from *Q. acutissima* mycorrhized with *T. indicum*. B, the ectomycorrhizosphere soil of *Q. acutissima*. C, the rhizosphere soil without *T. indicum* associations (control soil). D, roots from cultivated *Q. acutissima* without *T. indicum* colonization (control roots). Each value is the mean of three replicates (±SD). Values followed by different lowercase letters indicate significant differences ($P < 0.05$) between samples in a line.

the seedlings ($P < 0.05$). Inoculation with *T. indicum* thus clearly influenced the bacterial richness and diversity in the root endosphere and rhizosphere soil.

## Alpha diversity of fungal community

A total of 413,105 sequences were obtained from the 12 samples after quality control procedures, and there were 32,024–36,641 high-quality sequences in each sample (Fig. S2B). These high-quality sequences were clustered into 37–181 OTUs, representing five phyla, 13 classes, 33 orders, 59 families, and 89 genera. The Venn diagram is shown in Fig. 3B.

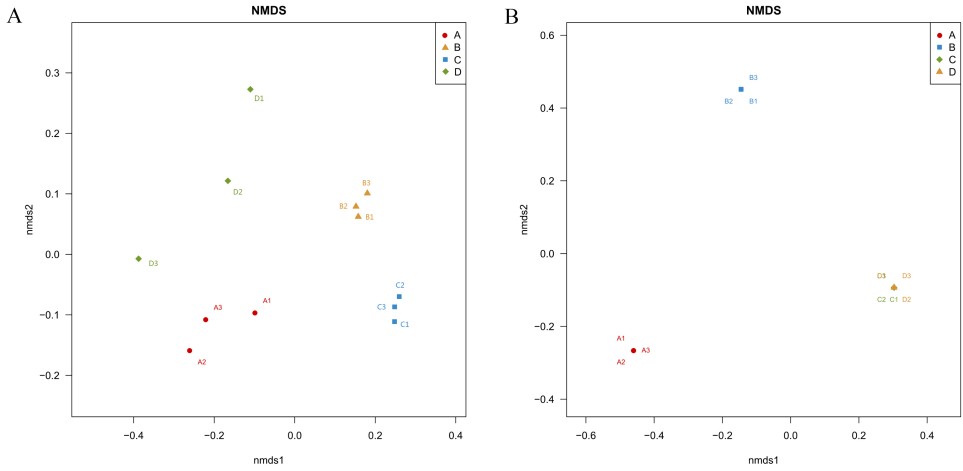

**Figure 4** **Nonmetric Multidimensional Scaling ordination showing the weighted UniFrac dissimilarities of bacterial (A) and fungal (B) communities in the roots and soil with *T. indicum* associations or not.** (A), ectomycorrhizae from *Q. acutissima* mycorrhized with *T. indicum*. (B), the ectomycorrhizosphere soil of *Q. acutissima*. (C), the rhizosphere soil without *T. indicum* associations (control soil). (D), roots from cultivated *Q. acutissima* without *T. indicum* colonization (control roots).

The number of observed species (OTU) differed significantly between the four treatments and was also significantly higher in the uninoculated rhizosphere soil ($P < 0.05$) (Table 3B). The Chao1 index differed significantly between the control treatments and the treatments associated with *T. indicum* ($P < 0.05$). The OTU and Chao1 indices indicated that the fungal community richness was significantly higher in the treatments without *T. indicum* colonization ($P < 0.05$). Inoculation with *T. indicum* decreased the fungal richness. Significant differences in the Shannon and Simpson indices were also observed between the uninoculated and inoculated samples ($P < 0.05$). The indices demonstrated that inoculation with *T. indicum* had an obvious impact on the fungal diversity, and *T. indicum* significantly decreased the fungal diversity in the root endosphere and rhizosphere soil ($P < 0.05$).

## Bacterial and fungal Unifrac-NMDS analysis

Unifrac-NMDS analysis was used to visualize the beta diversity of the bacterial and fungal communities in the different samples, which presents the similarities and differences in bacteria and fungi between the samples (Fig. 4). There were significant differences in bacterial community structure and composition between the four treatments (ANOSIM: $R > 0.9$, $P = 0.001$) (Fig. 4A).

The fungal community structures were very similar between the control root endosphere and control soil, and differed significantly from the treatments associated with *T. indicum* inoculation (ANOSIM: $R > 0.9$, $P = 0.001$) (Fig. 4B). Thus, inoculation with truffles altered the fungal community composition in the root endosphere and rhizosphere soil.
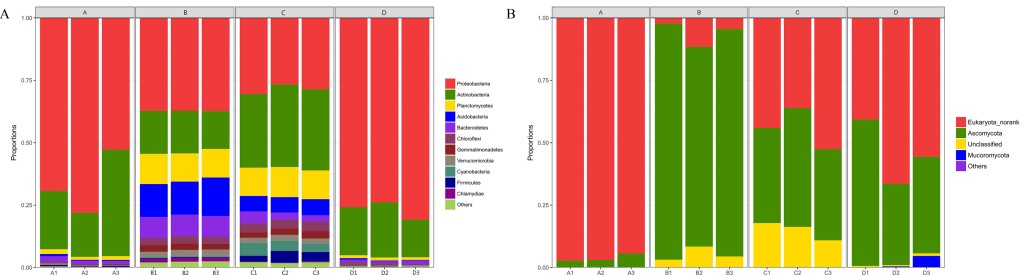

**Figure 5** **Taxonomic composition of bacterial (A) and fungal (B) communities associated with** ***Q. acutissima*** **root tips and rhizosphere soils at the phylum levels.** In the figure, 'A' indicates ectomycorrhizae from *Q. acutissima* mycorrhized with *T. indicum*; 'B', the ectomycorrhizosphere soil of *Q. acutissima*. 'C'; the rhizosphere soil without *T. indicum* associations (control soil); 'D', roots from cultivated *Q. acutissima* without *T. indicum* colonization (control roots); "norank" means there is no scientific name for this hierarchy in the taxonomic family tree; "Unclassified" means under the confidence threshold, it cannot be compared with the database. All experiments were conducted with three replicates.

## Taxonomic composition of the bacterial and fungal communities

The composition of bacterial communities at the phylum level was shown in Fig. 5A. A total of 30 phyla were identified, but only 14 phyla were detected in all 12 samples. Proteobacteria (52.4%), Actinobacteria (23.7%), Planctomycetes (6.5%), Acidobacteria (5.3%), Bacteroidetes (3.7%), and Chloroflexi (1.8%) were the six dominant phyla in all samples, accounting for 93.4% (average relative abundance) of all bacteria communities. At the class level, Proteobacteria_Unclassified (25.2%), Alphaproteobacteria (16.4%), Actinobacteria (16.0%), Gammaproteobacteria (7.6%), and Thermoleophilia (5.2%) were most abundant, all of which belong to Proteobacteria or Actinobacteria. At the genus level (Fig. S3A), the dominant genera included *Proteobacteria_Unclassified* (25.2%), *Streptomyces* (8.6%), *Pantoea* (5.3%), *Patulibacter* (2.3%), *SM1A02* (2.2%), and *Elev-16S-1332_norank* (1.9%).

The fungal community structure composition at the phylum level was shown in Fig. 5B. There were a total of five phyla in all the samples, including Ascomycota, Basidiomycota, Eukaryota_norank, Mucoromycota, and Unclassified. Among them, Ascomycota accounted for the relative abundance of 44.09%. At the class level, Eukaryota_norank (50.19%), Pezizomycetes (22.19%), and Sordariomycetes (17.14%) were the three most dominant. In addition, Dothideomycetes had a relative abundance of 4.10%. At the genus level (Fig. S3B), only six common genera existed in each sample of a total of 89 genera. The second-most abundant observed genus was *Tuber* (21.58%), and other genera with high relative abundances included *Humicola* (7.10%), *Fusarium* (5.81%), *Pleosporales_norank* (3.20%), *Collariella* (1.56%), and *Trichoderma* (0.72%).

## Differential analysis of bacterial and fungal communities

The abundances of some bacterial and fungal communities differed in the different samples. In terms of bacteria (Fig. S4A), the most abundant phylum, Proteobacteria, was significantly more abundant in the ectomycorrhizosphere soil than in the control soil ($P < 0.05$), whereas Proteobacteria was more abundant in the control root endosphere.

On the contrary, the control soil and ectomycorrhizae contained more Actinobacteria than the ectomycorrhizosphere soil ($P < 0.05$) and control roots, respectively. Acidobacteria and Bacteroidetes were significantly more abundant in the ectomycorrhizosphere soil ($P < 0.05$), and were more abundant in the inoculated treatments than in the control treatments. At the class level (Fig. 6A), Alphaproteobacteria, Betaproteobacteria, Deltaproteobacteria, and Acidimicrobiia had significantly higher abundance in the ectomycorrhizosphere soil ($P < 0.05$), while Thermoleophilia and Planctomycetacia were significantly more abundant in the control soil ($P < 0.05$). The ectomycorrhizae contained more Gammaproteobacteria than the control roots ($P < 0.05$). At the genus level (Fig. 7A), the samples associated with *T. indicum* inoculation had more *Streptomyces* and *SM1A02* than the control treatments, and the differences reached a significant level in soil samples ($P < 0.05$). *Kribbella* and *Variibacter* were more abundant in the ectomycorrhizae than in the control roots ($P < 0.05$), while *Patulibacter, Kribbella, Variibacter, Planctomyces,* and *Elev-16S-1332_norank* were more abundant in the control soil than in the ectomycorrhizosphere soil ($P < 0.05$). The ectomycorrhizosphere soil contained significantly more *Subgroup 7_norank, Rhizomicrobium,* and *Cytophagaceae_uncultured* than the control soil ($P < 0.05$).

In terms of fungi (Fig. S4B), ectomycorrhizosphere soil contained significantly more Ascomycota ($P < 0.05$), and its abundance was lowest in the ectomycorrhizae. Pezizomycetes was most abundant in the ectomycorrhizosphere soil (Fig. 6B), and its relative abundance was higher in the inoculated treatments than the inoculated treatments. The abundance of Sordariomycetes was higher in the treatments without *T. indicum* inoculation ($P < 0.05$). At the genus level (Fig. 7B), *Tuber* was only detected in the ectomycorrhizae and ectomycorrhizosphere soil with an average relative abundance of 43.11%, and was significantly more abundant in the ectomycorrhizosphere soil ($P < 0.05$). The abundances of *Humicola, Fusarium, Collariella,* and *Trichoderma* were significantly higher in the control samples ($P < 0.05$).

## DISCUSSION

As ectomycorrhizal fungi, truffles have an important function in the ecosystem. Ectomycorrhizal fungal communities can alter forest soil biogeochemistry, as observed in *T. melanosporum* and *T. aestivum* (Fu et al., 2016). However, whether the truffles can colonize the root tips of plants and successfully form ectomycorrhizae is not only determined by the surrounding abiotic factors, like the temperature and humidity, the soil pH and soil fertility, but also determined by biotic factors, such as soil microorganisms and vegetation (Slankis, 1974). In this study, we thus inoculated *T. indicum* on *Q. acutissima* to analyze how this affects the physiology of the host plant and shapes the soil properties and microbial communities in the root endosphere and rhizosphere soil during the early stage of symbiosis.

Ectomycorrhizae are beneficial to host plants in terms of water and nutrient uptake and host resistance (Harley & Smith, 2008). Previous studies showed that the symbiosis of *T. melanosporum* with *Pinus halepensis* seedlings improved the absorption of nutrient

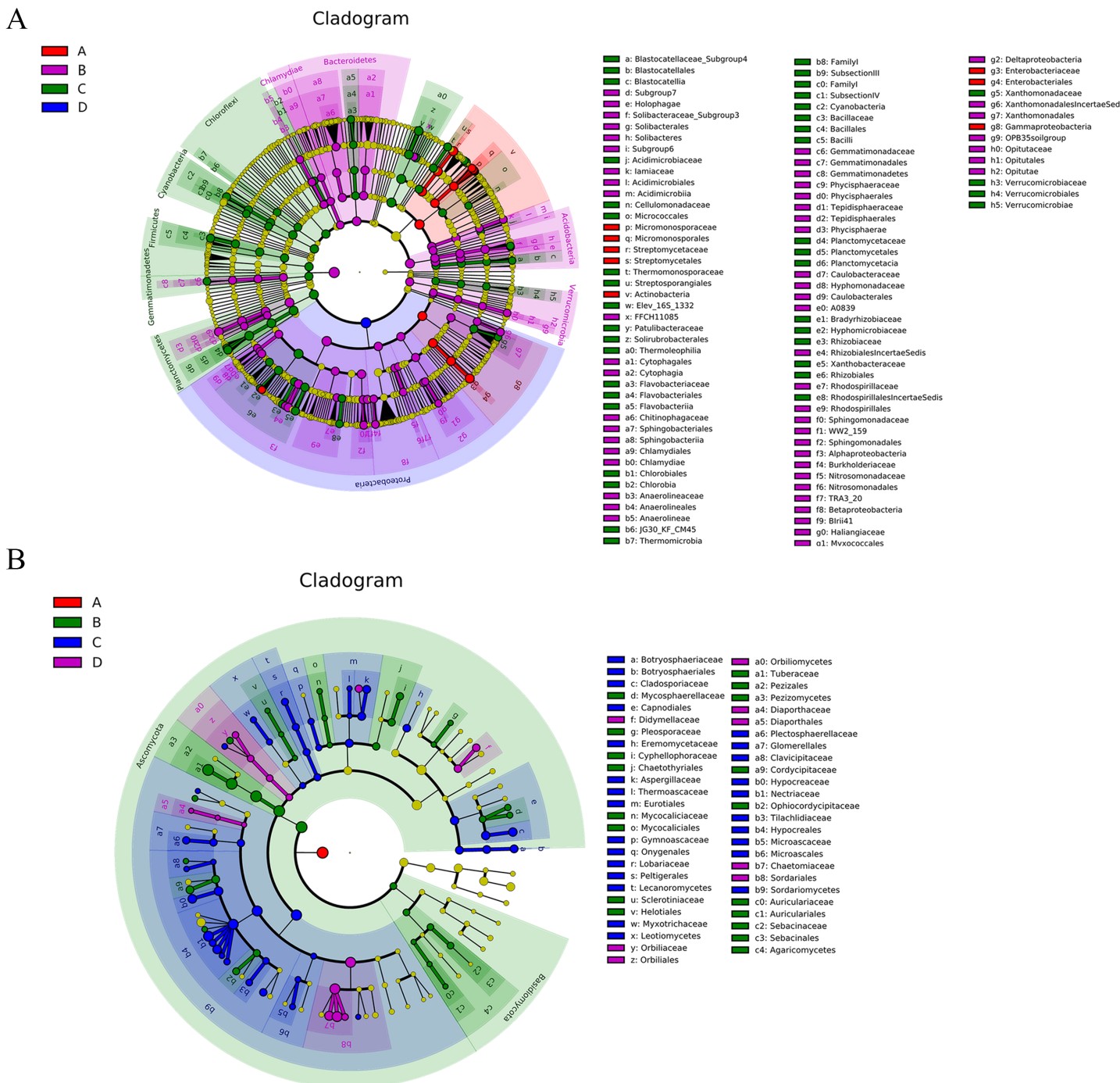

**Figure 6** The Cladogram based on LEfSe analysis ($P < 0.05$, LDA score $> 2$) showing the significantly differentially abundant bacterial (A) and fungal (B) taxa in the roots and soil with *T. indicum* inoculation treatments or not. In the key, 'A' indicates ectomycorrhizae from *Q. acutissima* mycorrhized with *T. indicum*; 'B', the ectomycorrhizosphere soil of *Q. acutissima*; 'C', the rhizosphere soil without *T. indicum* associations (control soil); 'D', roots from cultivated *Q. acutissima* without *T. indicum* colonization (control roots).

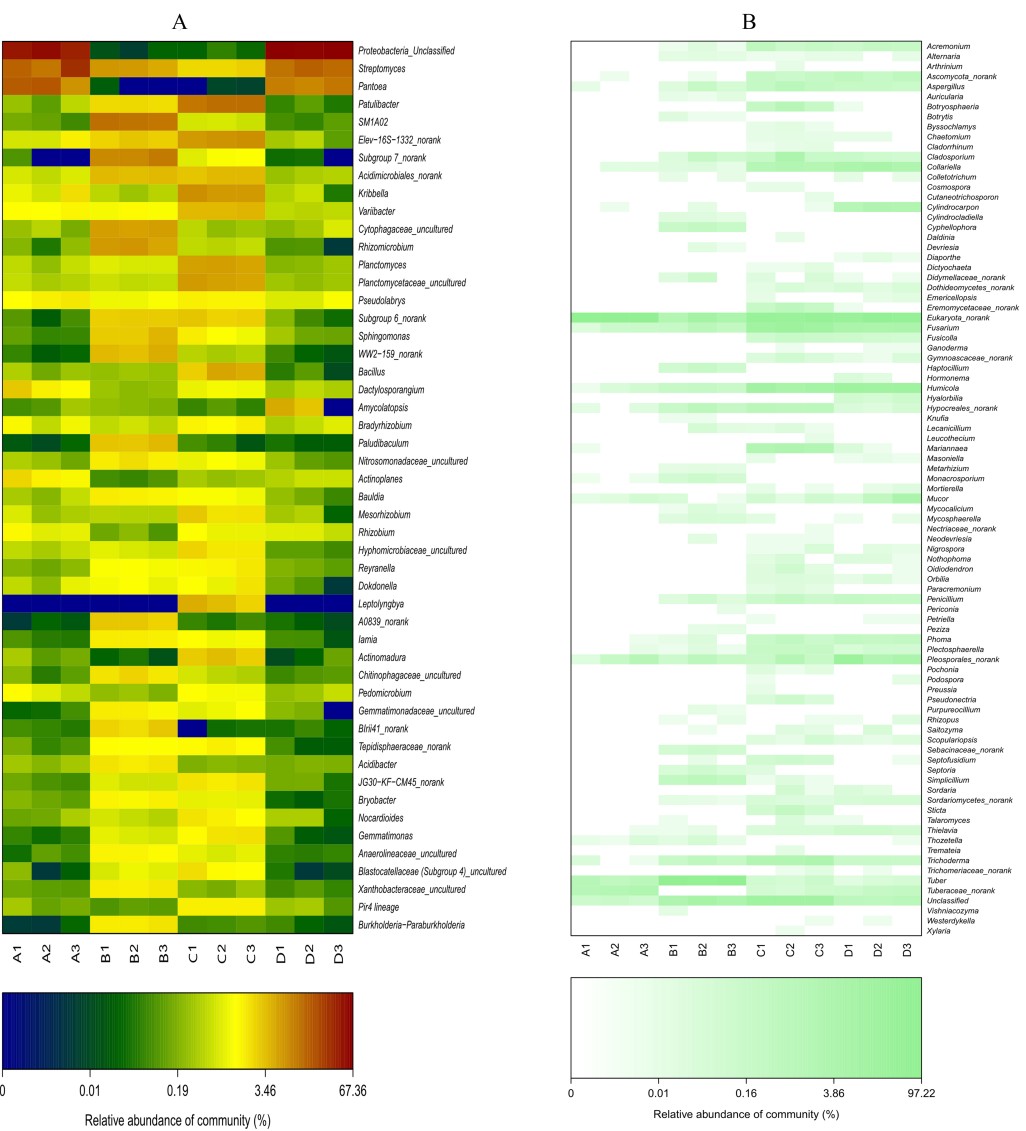

**Figure 7** **Heat-map analysis of the 50 most abundant bacterial (A) and fungal (B) genera in roots and soil communities of *Q. acutissima*.** In the figure, 'A' indicates ectomycorrhizae from *Q. acutissima* mycorrhized with *T. indicum*; 'B', the ectomycorrhizosphere soil of *Q. acutissima*; 'C', the rhizosphere soil without *T. indicum* associations (control soil); 'D', roots from cultivated *Q. acutissima* without *T. indicum* colonization (control roots). The relative abundances of OTUs at the genus level are colored according to the color scales at the bottom of each heatmap.

and the growth of host plants (*Dominguez et al., 2012*). Another study indicated that the inoculation of *T. indicum* on several Chinese indigenous trees led to higher average plant heights, higher ground diameter increments, and higher disease resistance and biomass compared with the seedlings that were not inoculated (*Hu, 2004*). Plant growth was found to be closely related to the measured physiological indicators in our study. We found that SOD activity increased significantly in the inoculated roots compared with the uninoculated roots, whereas the root activity decreased significantly. However, POD

activity, biomass and chlorophyll content didn't have significantly differences between the two different treatments. As the signaling compounds in transduction pathways, SODs play a very important role in plant physiology and act as inducers of cell damage when excessive production occurs at high concentrations (*Che et al., 2016*; *Río et al., 2018*). Increased SOD activity could be a response to abiotic stress in plants (*Ding et al., 2017*). The increase in SOD activity indicated that *T. indicum* inoculation improved the ability of the host plants to cope with stress. It is possible that during the early stage of symbiosis, the colonization of the seedling roots by the truffle hyphae to form ectomycorrhizae influences the environment around the roots, resulting in an increase in the metabolic activity of some substances in the roots, thereby altering the balance of active oxygen metabolism and causing increased SOD activity. Previous research showed that the root activity of *Pinus massoniana* inoculated with *T. indicum* was slightly higher compared with that of the seedlings without *T. indicum* colonization, which differed from our study (*Yin & Zhu, 2008*). Why *T. indicum* colonization decreased the root activity of *Q. acutissima* seedlings requires further study.

Soil parameter and nutrient availability dynamics might be crucial for supporting the lifecycle of truffle (*Marjanović et al., 2015*). The synthesis of truffle ectomycorrhizae was possibly related to the different soil properties (*Garcíamontero et al., 2006*). Our results showed that the ectomycorrhization of *Q. acutissima* with *T. indicum* affected the rhizosphere soil characteristics. Specifically, the organic matter, total nitrogen and available nitrogen, and total phosphorus were significantly higher in the ectomycorrhizosphere, indicating that the ectomycorrhizae of *T. indicum* promoted the release of nitrogen and phosphorus in the soil, particularly nitrogen, which would strengthen the absorption of these elements by the plant. A previous study showed that the formation of mycorrhizae of *Q. petraea*. with *T. melanosporum* increased the nitrogen content in plants, and the mycorrhization of *Pinus halepensis* Mill. with *T. melanosporum* improved the phosphorus uptake (*Domínguez Núñez, Planelles González & Rodríguez Barreal, 2008*). Other studies also confirmed that ectomycorrhizal fungi have the ability to improve nitrogen content (*Chalot & Brun, 1998*; *Dearnaley & Cameron, 2017*). However, the available phosphorus, available potassium, and available calcium were lower in the ectomycorrhizosphere, which differs from previous findings (*Fu et al., 2016*; *Li et al., 2017*). It is possible that these soil nutrients are relatively susceptible to the different conditions and different species. When establishing a new plantation for truffle, we should take consideration of some soil chemical properties including carbonates and pH (*Valverde-Asenjo et al., 2009*). *T. indicum* probably would like the slightly acidic soil, while *T. melanosporum* and *T. magnatum* prefer alkaline soil (*Li et al., 2017*).

The role of microorganisms in truffle formation is not to be underestimated and this may related to the changes of soil properties affected by truffle (*Ge et al., 2017*). Rhizosphere microbe communities can regulate mycorrhizal synthesis, and a symbiosis exists between some bacteria in the rhizosphere and the ectomycorrhizae. While these microbial communities promote the synthesis of ectomycorrhizae, the ectomycorrhizal fungi can also regulate the bacterial composition and function (*Song, Deng & Song, 2016*). In our results, the colonization of *T. indicum* reduced the bacterial richness, fungal richness

and fungal diversity in both the root endosphere and rhizosphere soil. Additionally, *T. indicum* colonization decreased the bacterial diversity in the rhizosphere soil but increased the bacterial diversity in the ectomycorrhizae. In addition, inoculation with *T. indicum* changed the composition of bacteria and fungus in the root endosphere and rhizosphere soil. Previous studies has provided evidence that neither in wild nor cultivated conditions, the diversity of bacterial and fungi reduced in the brûlé area as a result of truffles ensuring their survival by spreading their metabolites during their lifecycle (*Mello et al., 2013*). Interestingly, a previous report suggested that the volatiles released by truffles could inhibit both host and nonhost plants (*Splivallo et al., 2007*). Proteobacteria was found to be the most abundant bacteria in this study, and ectomycorrhizosphere soil and control roots contained more Proteobacteria. Many studies also showed that Proteobacteria was the dominant phylum in truffles fruiting bodies and in the different types of soils associated with *Tuber* (*Gryndler et al., 2013*; *Antony-Babu et al., 2013*; *Ye et al., 2018*). We also found that α-Proteobacteria was most abundant in the ectomycorrhizosphere soil, and γ-Proteobacteria was most abundant in the ectomycorrhizae. Some reports indicated that α-Proteobacteria and γ-Proteobacteria were the predominant truffle bacterial communities (*Barbieri et al., 2007*), which corroborates our results. In addition to Proteobacteria, we also detected Actinobacteria, Planctomycetes, and Acidobacteria to be dominant bacteria. *Streptomyces* and *SM1A02* may play important roles in the mycorrhization of *Q. acutissima* with *T. indicum,* as the treatment associated with *T. indicum* had significantly more *Streptomyces* and *SM1A02. Streptomyces* was also detected in the ectomycorrhizae of *T. aestivum* in a previous study (*Gryndler & Hršelová, 2012*). In addition, the ectomycorrhizosphere soil contained significantly more *Rhizomicrobium*. In terms of fungus, *Tuber* was the second-most abundant fungus, accounting for 43.11% of the fungal community in the ectomycorrhizae and ectomycorrhizosphere soil. Some species of *Fusarium* are pathogenic fungi that produce fungal toxins (*Gerlach & Nirenberg, 1982*). Our study found a lower abundance of *Fusarium* in the ectomycorrhizae and ectomycorrhizosphere soil, indicating that *T. indicum* inoculation reduced some pathogenic fungi. Previous work suggested that ectomycorrhizal fungi could protect the plant host during their growth, for example, they could reduce the infection of plants by other microbes (*Kennedy, 2010*). Overall, the microbe communities in the soil, ectomycorrhizae, and fruiting bodies, including bacteria, yeast, and filamentous fungi, are important and closely related to the development and nutrition of truffle ascocarps (*Splivallo et al., 2015*). Rhizosphere effects are huge on the root-associated microbial community structure, which was an important reason why the root endophere microbial community differed from rhizosophere community of *Q. acutissima* plants. Plants can induce or shape rhizosphere microflora through their roots' secretion, and the pants genotype and soil type are the important driving forces to microbial community structure (*Berendsen, Pieterse & Bakker, 2012*; *Doornbos, Loon & Bakker, 2012*), which may be related to the mycorrhizae formation. Furthermore, there is some interaction between the soil communities and endophytes. It was reported that both the internal and external parts of truffles could be colonized by bacteria, and these communities were probably from the rhizosphere soil at the early stage of truffle formation (*Antony-Babu et al., 2013*). Although the microbial diversity decreased, there

were also some communities that showed a higher abundance in the ectomycorrhizae and ectomycorrhizosphere. These communities were thus closely related to the fomation of truffle mycorrhizae, possibly playing a role in ectomycorrhizal synthesis. However, the specific interactions between these microbial communities, ectomycorrhizosphere soil properties and truffles require further exploration.

## CONCLUSIONS

This study revealed a significant effect of *T. indicum* inoculation on the *Q. acutissima* seedlings, the rhizosphere soil properties, and the microbial communities (bacteria and fungus) in root endosphere and rhizospher soil, which gave more details to the symbiosis of *Q. acutissima* with *T. indicum*. During the early symbiotic stage, changes in SOD activity, root activity and some rhizosphere soil properties (e.g., organic matter, total nitrogen, and available nitrogen) due to the *T. indicum* colonization may be the important factor that drove the changes in microbial communities. It was clear that the *T. indicum* inoculation did alter the diversity and structure of the microorganisms in root endosphere and rhizosphere soil, and certain bacteria and fungi were greatly different. Their specific roles playing in the truffle growth remained to be disclosed.

## ACKNOWLEDGEMENTS

We would like to thank LetPub for providing linguistic assistance during the preparation of this manuscript.

### Funding

This work was supported by the Science and Technology Support Project in Sichuan Province (2016NYZ0040) and the Sichuan Mushroom Innovation Team. The funders had no role in study design, data collection and analysis, decision to publish, or preparation of the manuscript.

### Grant Disclosures

The following grant information was disclosed by the authors:
Science and Technology Support Project in Sichuan Province: 2016NYZ0040.
Sichuan Mushroom Innovation Team.

### Competing Interests

The authors declare there are no competing interests.

### Author Contributions

- Xiaoping Zhang performed the experiments, analyzed the data, prepared figures and/or tables, authored or reviewed drafts of the paper.
- Lei Ye performed the experiments, analyzed the data, contributed reagents/materials/-analysis tools, prepared figures and/or tables.

- Zongjing Kang performed the experiments, contributed reagents/materials/analysis tools.
- Jie Zou performed the experiments, analyzed the data.
- Xiaoping Zhang conceived and designed the experiments, approved the final draft.
- Xiaolin Li conceived and designed the experiments, authored or reviewed drafts of the paper, approved the final draft.

## Data Availability

NCBI accession numbers: SRR7791517–SRR7791532 in BioProject ID PRJNA489558.

## Supplemental Information

Supplemental information for this article can be found online at http://dx.doi.org/10.7717/peerj.6421#supplemental-information.

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
