# Peer review of "Mycorrhization of Quercus acutissima with Chinese black truffle significantly altered the host physiology and root-associated microbiomes"

_PeerJ, doi:10.7717/peerj.6421_

## Round 0.1 · original submission · Minor Revisions

Both reviewers and myself consider your work interesting deserving publication. However there are some aspects that need clarification and some rewording. I invite you to take into account all of them.

·

Basic reporting

In this study, Zhang et al., evaluated how the ectomycorrhizae of Quercus acutissima with Chinese black truffle affects plant physiology using four parameters. Further, they collected the root endosphere and rhizosphere samples of those T.indicum inoculated and uninoculated plants, and profiled the bacterial and fungal communities in these samples. The methodologies used in this study could be strong enough to answer their questions, and the overall quality of the writing is OK despite some places being vague in delivering their meanings. I suggest the authors doing some changes before publishing this work, and details are provided as follows.

Experimental design

The general experimental design was good enough to answer research question of the authors.

Validity of the findings

the general bioinformatics and statistical analysis were OK, but need to be improved at some places please see detail in my general comments for the authors

Additional comments

Comments on Peer J paper

In this study, Zhang et al., evaluated how the ectomycorrhizae of Quercus acutissima with Chinese black truffle affects plant physiology using four parameters. Further, they collected the root endosphere and rhizosphere samples of those T.indicum inoculated and uninoculated plants, and profiled the bacterial and fungal communities in these samples. The methodologies used in this study could be strong enough to answer their questions, and the overall quality of the writing is OK despite some places being vague in delivering their meanings. I suggest the authors doing some changes before publishing this work, and details are provided as follows.

Title: I may suggest to change the current title according to your main findings, for example: Mycorrhization of Quercus acutissima with Chinese black truffle significantly altered host physiology and root-associated microbiomes.

In abstract, Line 37: throughout the article, I could see many phrases of ‘surrounding soil’, did you mean ‘rhizosphere soil’. Generally, rhizosphere effects are huge on soil microbial properties. I suggest to make it clear by using rhizosphere/root endosphere throughout this manuscript.

Introduction: I suggest to provide hypothesis that you were testing in this study at the end of the introduction.
Line 119, what was the size of your plastic containers
Line 130, it would benefit if providing more details about how the glasshouse conditions were controlled, e.g., temperature, moisture, and what time of the year as it matters daylight.
Line 137: please detail what soil properties you measured.
Line 140, how you cleaned the roots?
Line 144, it is important to provide more details how the mycorrhizal colonization rating was performed.

Bioreplicates: you had 60 plants for each treatment, thereby you would have 20 plants per each bilreplicate, please provide relative information at an appropriate place of the manuscript

Line 173 ‘soil samples around roots’, did you mean rhizosphere soils?

Line 177, please cite references that originally designed the primer pairs for bacterial and fungal community profiling you used here, and provide template specific primer sequences.

Line 194, which version of the SILVA and Unite databases?

Line 203, regarding statistics of the study: how you performed multivariate analysis for your community data? And in the following results, you should also provide P values demonstrating if your treatments differed from controls.
Line 209, did you collect the plant biomass production data?

Line 212, should you explain these mycorrhizae-induced differences in morphology in more details? As I see there seems to be a lot of differences between inoculated and procedure control roots in Figure 1.

Line 222, soil properties analysis, should you give the size of changes in particular soil nutrient? such as total nitrogen (+110%)…..

Line 235 Alpha diversity of bacterial community
Line 258 Alpha diversity of fungal community
Line 243-257 and Line 263-271, if possible, provide significant level for different treatments, P<0.01 or P<0.05.

Line 272, similar to the above, statistical analysis here needs to be improved.

Line 336, typo-mistakes

Line 403, typo-mistakes

Line 412: not all Fusarium species are phytopathogens

In discussion, if appropriate, you can discuss more about rhizosphere effects on root microbial community structure, that is why the endophere microbial community differed from rhizosophere community for Q. acutissima plants.

In all figures, should clarify what treatments associate with A, B, C and D, figures should be shelf-explanative.
Figure 5, it might be better if only show differences at phylum level, results at genus level can be therefore put in Supplementary results.

·

Basic reporting

no comment

Experimental design

no comment

Validity of the findings

no comment

Additional comments

The manuscript described an original work on the impact of truffle on root/soil characteristics. Experimental design seems to be right with respect to the aims of the performed work. Results seem to be solid basing on three replicates. However, some points have to be clarify before the publication.
In the introduction the part from line 54 to line 56 should be deleted, since it is not well inserted there.
Line 62, colonization should be better than infection.
Line 148-51, why ECM, soil, soil, roots and not ECM, soil, roots, soil?
Root activity and chlorophyll content. Why the authors have evaluated these parameters, and mainly why chlorophyll content in roots?
The sentence from line 162 to line 163 have to be clarified.
Line 367: why "could"?
In Figure 2, microstructure characteristics is not suitable...morphological characteristics could be probably better. Legend should be changed in:
a and c, transversal sections; b and d, longitudinal sections.
Check along the manuscript "T.indicum" that should be "T. indicum" and "Q.acutissima" should be "Q. acutissima". The same comment also for T. melanosporum.
Recent literature has been cited, although the authors should be check the word that need italic (such as Tuber melanosporum) in the reference list.

---

## Round 0.2 · accepted · Accept

Thanks for taking into consideration final comments made by the reviewers. Your manuscript is ready for publication.

# ·

Basic reporting

I believe the current version meet publishing standard.

Experimental design

I believe the current version meet publishing standard.

Validity of the findings

I believe the current version meet publishing standard.

Additional comments

I believe the current version meet publishing standard.

·

Basic reporting

The manuscript has been improved following the reviewers' suggestions. I have only a doubt on the fact to use microbiomes or microbiota in the title. I prefer the second one, but see also what the second reviewer will say on this point.

Experimental design

Methods have been improved in the revised version.

Validity of the findings

In the revised version the critical points have been addressed.

Additional comments

The manuscript is now suitable for the publication.